# Regulation of Cyclooxygenase-2 Expression in Human T Cells by Glucocorticoid Receptor-Mediated Transrepression of Nuclear Factor of Activated T Cells

**DOI:** 10.3390/ijms232113275

**Published:** 2022-10-31

**Authors:** Cristina Cacheiro-Llaguno, Elena Hernández-Subirá, Manuel D. Díaz-Muñoz, Manuel Fresno, Juan M. Serrador, Miguel A. Íñiguez

**Affiliations:** 1R&D Unit Allergy & Immunology, LETI Pharma, S.L.U., Tres Cantos, 28760 Madrid, Spain; 2Labcorp Development S.A.U., 28050 Madrid, Spain; 3Toulouse Institute for Infectious and Inflammatory Diseases, Inserm, CNRS, University Paul Sabatier, CHU Purpan, 31300 Toulouse, France; 4Departamento de Biología Molecular and Instituto de Biología Molecular (IUBM), Universidad Autónoma de Madrid (UAM), 28049 Madrid, Spain; 5Instituto de Investigación Sanitaria La Princesa, 28006 Madrid, Spain; 6Immune System Development and Function Unit, Centro de Biología Molecular “Severo Ochoa” (CBMSO), Consejo Superior de Investigaciones Científicas (CSIC)-UAM, 28049 Madrid, Spain

**Keywords:** glucocorticoids, glucocorticoid receptor, transrepression, Cyclooxygenase-2, T cells, NFAT

## Abstract

Cyclooxygenase (COX) is the key enzyme in prostanoid synthesis from arachidonic acid (AA). Two isoforms, named COX-1 and COX-2, are expressed in mammalian tissues. The expression of COX-2 isoform is induced by several stimuli including cytokines and mitogens, and this induction is inhibited by glucocorticoids (GCs). We have previously shown that the transcriptional induction of COX-2 occurs early after T cell receptor (TCR) triggering, suggesting functional implications of this enzyme in T cell activation. Here, we show that dexamethasone (Dex) inhibits nuclear factor of activated T cells (NFAT)-mediated COX-2 transcriptional induction upon T cell activation. This effect is dependent on the presence of the GC receptor (GR), but independent of a functional DNA binding domain, as the activation-deficient GRLS7 mutant was as effective as the wild-type GR in the repression of NFAT-dependent transcription. Dex treatment did not disturb NFAT dephosphorylation, but interfered with activation mediated by the N-terminal transactivation domain (TAD) of NFAT, thus pointing to a negative cross-talk between GR and NFAT at the nuclear level. These results unveil the ability of GCs to interfere with NFAT activation and the induction of pro-inflammatory genes such as COX-2, and explain some of their immunomodulatory properties in activated human T cells.

## 1. Introduction

Prostaglandin (PG) H endoperoxide synthase, or cyclooxygenase (COX), catalyzes the two-step conversion of AA to PGH_2_, the first reaction required for the biosynthesis of PGs and thromboxanes. At least two isoforms of the enzyme are expressed in mammalian tissues, COX-1 and COX-2 [1,2,3]. COX-1 is constitutively expressed in most tissues, and is thought to be involved in homeostatic prostanoid biosynthesis. In contrast, COX-2 is induced by various pro-inflammatory agents, including cytokines and mitogens. COX-2 is considered the predominant isoform involved in the inflammatory response. Accordingly, the ability of non-steroidal anti-inflammatory drugs (NSAIDs) to inhibit COX-2 activity may explain their therapeutic effects as anti-inflammatory drugs, whereas the inhibition of COX-1 activity may account for some of their unwanted side effects [4,5,6]. Therefore, most of the new research on anti-inflammatory drugs has been aimed at targeting the COX-2-inducible production of PGs, not only at the enzymatic level, but also as regulators of its transcriptional induction. Different transcription factors can regulate COX-2 transcription depending on the stimulus and cell type. The human COX-2 gene promoter contains binding sites for transcription factors such as nuclear factor (NF)-κB, NF-IL6/CCAAT enhancer binding protein and cAMP response element binding protein (CREB). These sequences are positive regulatory elements for the transcription of COX-2 expression in different cell types, including vascular endothelial cells, colon carcinoma cells, and monocytes [7,8,9,10,11]. Moreover, we have previously reported that the transcriptional induction of COX-2 occurs as early as 1 h after T cell receptor triggering, suggesting functional implications of COX-2 activity in T cell activation process. COX-2 expression was induced upon T cell activation in a cyclosporin A (CsA)-sensitive manner [12]. A detailed analysis of the COX-2 promoter region revealed the presence of two NFAT binding sites required for the induction of COX-2 promoter activity in T cells [13] that have also been demonstrated to be essential in COX-2 transcriptional induction in other cell types and upon a variety of stimuli [14,15,16,17,18,19].

One of the best-known anti-inflammatory drugs regulating the transcriptional induction of inflammatory mediators, including COX-2, are glucocorticoids (GCs). These agents display potent immunomodulatory and anti-inflammatory properties, being widely used as therapeutic agents to treat a broad range of autoimmune and chronic inflammatory diseases [20]. Regarding T cells, the most recognized biologic effect of GCs on peripheral T cells is immunosuppression, which is due to the inhibition of the expression of a wide variety of activation-induced pro-inflammatory genes, including tumor necrosis factor (TNF)-α and interleukins (IL)-2, -6, -1α, and -1β [21]. GCs mediate these biological effects through binding to an intracellular glucocorticoid receptor (GR). Upon ligand binding, the GR translocates to the nucleus, where it participates in the regulation of gene expression, both positively (transactivation) or negatively (transrepression) [22,23,24]. The transcriptional induction of gene expression by GCs depends on ligand-activated GR binding to glucocorticoid response elements (GRE) in the promoter region of target genes, as in the case of genes involved in glucose and fat metabolism [25,26]. On the other hand, several mechanisms have been suggested for the negative regulation of gene expression by GCs, such as the activation of GR-dependent inhibitory genes; binding to negative GREs; or, most commonly, by transcriptional interference involving competition with coactivators, as well as by GR interaction with transcription factors [22,27,28]. GR-mediated transcriptional repression (transrepression) is the main mechanism by which GCs inhibit the activity of several transcription factors, including NFκB, activator protein (AP)-1, CREB, signal transducers, and activators of transcription (STATs) or interferon-regulatory factors (IRFs), among others. The tethering of ligand-activated GR to these regulatory transcription factors involved in the upregulation of inflammatory genes is the main mechanism described to explain the anti-inflammatory and immunomodulatory properties of GCs. Therefore, GCs inhibit the synthesis, release, and/or action of cytokines and other mediators that promote inflammatory and immune responses [20,22,29]. In this regard, it is well known that COX-2 expression is negatively regulated by GCs such as dexamethasone (Dex) in numerous types of cells, both at the transcriptional and post-transcriptional level [30,31,32,33,34,35]. Here, we have analyzed the effects of Dex on COX-2 expression in T cells. Our results show that GCs inhibit NFAT-mediated COX-2 transcriptional induction in activated T cells. These effects were dependent on the presence of the GR, but were independent of its ability to bind DNA. These results provide new evidence about the anti-inflammatory and immunomodulatory properties of GCs, through their ability to interfere with the activation of NFAT and the induction of NFAT-dependent transcription of pro-inflammatory genes in activated human T cells.

## 2. Results

### 2.1. Inhibition of COX-2 Expression by Glucocorticoids

To explore the influence of GCs on COX-2 gene expression in activated T cells, we used a human T cell leukemia-derived Jurkat cell line, which does not express a functional endogenous GR (Jurkat), as well as two clones derived from this parental cell line upon stable transfection with expression vectors for a wt version of the rat GR (J-GRwt) or for the GR LS7 mutant (J-GRLS7), as previously described [36,37]. The LS7 mutant contains two adjacent two amino acid exchanges in the second zinc finger of the DNA binding domain (DBD). It has been reported to be a poor transactivator at GRE promoters, while retaining efficient activity in repressing AP-1- and NF-κB-dependent transcription, comparable with that of wt GR [37]. Assays with parental Jurkat T cells transiently transfected with both GR expression vectors were also performed in order to validate the results obtained with stable transfectants.

First, we checked the ability of transfected GRwt to promote GC response element (GRE)-mediated transactivation by using a GRE-Luc reporter containing two copies of a consensus GRE present in the long terminal repeat (LTR) of the mouse mammary tumor virus (MMTV). Consistent with previous reports [37], whereas the treatment of parental Jurkat cells with the synthetic GC dexamethasone (Dex) did not result in a significant induction of GRE-mediated luciferase activity, cells stably or transiently transfected with GRwt demonstrated full Dex-dependent transactivation (Appendix A). On the other hand, cells bearing the LS7 GR mutant exhibited minimal GRE-dependent transactivating potential.

We analyzed the effect of Dex on COX-2 expression in purified human T cells, as well as on the different subclones of Jurkat cells, upon stimulation. We have previously shown that COX-2 expression is induced in T cells by stimuli resembling T cell activation, such as TCR crosslinking or phorbol ester plus calcium ionophore (PMA + Ion) treatment [12]. In addition to its well-known inhibitory effects on induced IL2 expression, Dex treatment severely reduced the PMA+Ion- or antiCD3/CD28-mediated increase in COX-2 expression in purified human T cells both at the mRNA and protein level (Figure 1A,B). Moreover, this reduction in COX-2 expression upon Dex treatment resulted in a decrease of COX-2-mediated Prostaglandin (PG) production. As shown in Figure 1C, the stimulation of T cells with PMA + Ion promoted a COX-2 -dependent increase in PGE_2_ and PGF2_α_ synthesis that was abrogated by the COX-2 enzymatic inhibitor NS398. This increase in PG production in activated T cells was significantly reduced by Dex or CsA treatment as a consequence of the inhibitory effect of these agents on COX-2 expression.

To further explore the mechanism of action of GCs on T cells, we analyzed, by real-time RT-PCR, the effect of Dex on gene expression in parental and GR-transfected Jurkat T cells. PMA + Ion stimulation led to a substantial increase in COX-2 expression in these cells that was blocked in the presence of the immunosuppressor CsA, independently of GR expression. However, the quantification of the effects of Dex on mRNA levels in PMA + Ion treated Jurkat T cell clones demonstrated the GR dependence for the inhibitory actions of Dex (Figure 2A). Interestingly, Dex efficiently inhibited COX-2 (Figure 2A), as well as IL2 and TNFα (Appendix A) induced mRNA levels, in cells expressing the activation-deficient GR mutant LS7. Moreover, COX-2 protein induction after T cell activation was also clearly inhibited by Dex in the presence of GR, either wt or LS7 mutant (Figure 2B), thus confirming that Dex-GR mediated trans-repression is dependent on GR, but independent of GR binding to the DNA.

### 2.2. Glucocorticoids Inhibit COX-2 Promoter Activity

The effects of GCs on the transcriptional regulation of COX-2 were also tested by analyzing COX-2 promoter-driven transcription in transiently transfected Jurkat cells. The 5′ flanking region of the COX-2 gene contains an E-box, a CRE, and functional binding sites for NFκB. In addition, we have identified two NFAT sites involved in the regulation of COX-2 gene expression during T cell activation (Figure 3A). We conducted transfection experiments in Jurkat cells with Luc reporter plasmids driven by a series of deletion fragments spanning from positions −1796 bp to −46 bp of the COX-2 gene transcription start site. As expected, transcription driven by constructs with deletions spanning from −1796 to −170 of the COX-2 promoter region (P2-1900 and P2-274) was efficiently induced by PMA + Ion, but not the −46 to +104 bp promoter region (P2-150), in which the main regulatory sequences are absent. In accordance with Western blot and RT-PCR experiments, Dex inhibited the activation of COX-2 promoter P2-1900 and P2-274 constructs by PMA + Ion only in cells transiently transfected with expression vectors for the GRwt (pGRwt) or the mutant version (pGRLS7), but not in parental cells transfected with an empty vector (Figure 3B). Similar results were obtained with cells stably expressing GR constructs, J-GRwt, and J-GRLS7 (Figure 3C). We also analyzed Dex effects on a COX-2 promoter construct bearing a mutated NFκB site (P2-431 NFκBmut) in Jurkat, J-GRwt, and J-GRLS7 cells upon anti CD3/CD28 stimulation. As shown in Figure 3D, Dex efficiently inhibited anti CD3/CD28-mediated COX-2 induction in J-GRwt and J-GRLS7 cells in the absence of a functional NFκB site. Therefore, cis-acting elements mapping between the nucleotides at positions −170 to −46 appear to be required not only for the activation, but also for Dex inhibition of the COX-2 promoter. This region contains the proximal and distal NFAT sites essential for COX-2 promoter induction in activated T cells.

Analogous results were obtained in studies analyzing the actions of Dex on the transcriptional activation of IL2 and TNFα promoters (Appendix A), thus confirming that the transrepression exerted by Dex on the transcriptional activation of activated T cells is dependent of GR, but independent of GR binding to the DNA.

### 2.3. Inhibition of NFAT -Mediated Transactivation by Glucocorticoids

One of the main mechanisms of GR-mediated transcriptional repression is related to its interference with the function and activation of transcription factors, such as AP-1 or NFκB, via protein–protein interaction [21,38,39,40,41]. Interference with other transcription factors, such as NFAT, has been also reported, although how NFAT-dependent transactivation is inhibited by GCs remains unclear [42,43,44,45]. Transcriptional activation of a variety of cytokines, including TNFα and IL2, in immune cells, depends on the coordinate interactions among several transcription factors, including members of the NFAT family. NFAT also plays a pivotal role in the transcriptional activation of COX-2 not only in T cells, but also in a variety of cell types [14,15,16,17,18,19]. In order to determine whether the effects of Dex could be occurring through the inhibition of NFAT-mediated transactivation of the COX-2 gene, we tested the ability of GR-Dex to interfere with transactivation of NFAT-dependent reporter constructs. For this, parental or GR-transfected Jurkat cells were co-transfected with reporter constructs containing different cis-acting NFAT response elements. As shown in Figure 4A, NFAT-dependent transcription driven by NFAT sites present in the promoter of IL2 (pNFAT-Luc) was strongly induced upon cell activation with PMA + Ion. In both cell lines, bearing either GRwt or GRLS7, this induction was markedly suppressed in response to Dex. Similar results were obtained in cells stimulated with anti CD3/CD28 Abs (Figure 4B). We also examined the ability of Dex to repress NFAT luciferase reporter plasmids lacking NFAT:AP1 composite sites, such as those present in the IL13 or the IL4 promoters, whose induction relies on NFAT, but not on AP-1 [46,47,48] (Figure 4C). No considerable differences in the transrepression activity of the GR upon Dex treatment were observed among the different NFAT reporters assayed.

In resting T cells, NFAT is a cytoplasmic factor that, upon activation, is dephosphorylated by the calcineurin phosphatase and translocated into the nucleus, leading to the transactivation of target genes [49,50,51]. As shown in Figure 5, NFAT was dephosphorylated upon activation with PMA + Ion, showing a decrease in phosphorylated NFAT signal and an increase in a band of dephosphorylated NFAT, which migrates faster in SDS-PAGE electrophoresis. However, whereas CsA almost abolishes the dephosphorylation of NFAT in the Jurkat cell clones studied, Dex treatment did not change the profile of phosphorylated/dephosphorylated NFAT in control or PMA+Ion-treated Jurkat cells expressing GRs comparing to parental ones.

### 2.4. Overexpression of NFAT Restores Promoter Activity in the Presence of Dex

To test whether the GR was interfering with the ability of NFAT to transactivate genes, Jurkat cell lines were co-transfected with increasing quantities of an expression vector encoding NFATc2, and the activity of the COX-2 promoter was analyzed. As shown in Figure 6A, the overexpression of NFATc2 increased the PMA + Ion activation of COX-2 promoter in Jurkat cells. Interestingly, increased NFATc2 expression renders transcriptional activity resistant to inhibition by Dex in cells expressing either the GRwt or the GRLS7. Further evidence of the transcriptional interference between the GR and NFAT was shown from experiments where the transrepression by Dex of the NFAT-Luc reporter was analyzed. In this case, we determined the cooperation of the overexpression of NFATc2 with the treatment of PMA in the absence of Ca^++^ ionophore treatment for the induction of NFAT-Luc reporter activity. As shown in Figure 6B, PMA treatment in the presence of increasing quantities of transfected NFATc2 expression vector promoted an increase in NFAT-mediated transcription, clearly observed with the higher doses of vector transfected (0.5 and 1 µg). Noteworthy, Dex-mediated inhibition of NFAT-Luc activity was essentially abrogated in the case of cells cotransfected with GRwt when the highest dose of the expression vector pNFATc2 (1 µg) was used, and substantially reversed in the case of GRLS7. Altogether, these results suggest a negative cross-talk between GR and NFAT signaling that results in transcriptional interference in the regulation of NFAT-mediated gene expression in T cells.

### 2.5. Effects of Glucocorticoids on NFAT Transactivation Activity

Upon activation, nuclear NFAT is able to increase transcription of genes through still not well-understood mechanisms mediated by its transactivation domain in such a way that stimuli leading to NFAT translocation and DNA binding were also able to induce transactivation mediated by the N-terminal transactivation domain (TAD) of NFAT proteins [52,53]. Therefore, we tested whether GCs had an effect on this step in the NFAT signaling pathway. Jurkat T cells either stably or transiently transfected with GR constructs were co-transfected with a plasmid containing the transactivation domain of NFATc2 (1–415) fused to the GAL4 DNA binding domain (DBD), along with a GAL4-Luc reporter.

We measured the ability of PMA + Ion to increase the transactivation of GAL4-NFATc2 TAD in the absence or presence of Dex. In agreement with previous reports [54,55], PMA + Ion induced a 3–5-fold increase in the transactivating activity mediated by the GAL4-NFATc2 (1–415) construct. As shown in Figure 7A, Dex efficiently inhibited PMA+Ion-induced NFAT transactivating activity in cells transfected with the wt or the mutant GR expression vectors. Similar results were obtained with J-GRwt or J-GRmut clones when we analyzed the behavior of the GAL4-NFATc2 construct containing the entire TAD (1–415), along with a control construct bearing the first 104 aa of the NFATc2 TAD (1–104), which has been previously shown to be unresponsive to PMA + Ion (Figure 7B) [54].

## 3. Discussion

GCs are known to negatively regulate COX-2 expression, both at the transcriptional and post-transcriptional level, in several cell types, including cells of the immune system [30,31,32,33,34,35,56]. Although most of the studies carried out so far have been focused on the actions of GCs on cells of the innate immune response, studies of the effects exerted of GCs on T cells are also of great importance. Indeed, conditional mice deficient for GR in T cells are completely resistant to the anti-inflammatory actions of Dex in vivo [57]. Here, we have analyzed the effects of Dex, a synthetic GC, on the expression of COX-2 upon T cell activation. Dex treatment was as effective as CsA in the inhibition of the induction of COX-2 expression in primary human T cells. COX-2 mRNA induction and inhibition were paralleled by changes in COX-2 protein levels. Moreover, the inhibition of COX-2 expression by Dex resulted in decreased COX-2-mediated PGE_2_ production.

In order to determine the influence of GR in the inhibitory actions of Dex in T cells, we used a Jurkat cell line resistant to GCs, transiently or stably transfected with either a GRwt or with a transactivation-defective mutant of GR, which cannot bind DNA (GRLS7) nor drive Dex-mediated transactivation of gene expression, but which is still fully competent in transrepression [37]. The repression of PMA + Ion induction of COX-2 expression by Dex was dependent on the expression of the GR, either in its wild-type form or in its mutated version. Dex-mediated transrepression in cells expressing GRLS7 was as effective as those transfected with the GRwt, not only in the inhibition of COX-2 induction, but also in the repression of IL2 and TNFα expression. These results indicate that GR-mediated transrepression in T cells is independent of GR binding to DNA. This mode of action has been described to rely on the interference with the activity of transcription factors by protein–protein interaction with the GR. Accordingly, other transactivation-defective mutants of GR are also fully competent in transrepression, reducing the expression of AP-1 or NFκB driven genes in the absence of DNA binding [38,39,58]. Moreover, the relevance of the DNA binding-independent actions of GR has been also evidenced in vivo in mice that harbor the GR^dim^ mutation [59]. Several studies have reported protein–protein interaction between the GR and several transcription factors [42,60,61]. Interestingly, Chen, R. et al. [42] have described NFATc and GR co-precipitation in nuclear extracts of Dex-treated and calcium-stimulated T cells, which supports a model in which the inhibition of IL4 expression by GCs could take place through GR interference with NFATc binding to the IL4 promoter via direct protein–protein interaction.

The inhibitory effects of GCs on the promoter activity of COX-2, as well as on IL2 and TNFα promoters, suggested that GR-mediated repression occurs mainly at the transcriptional level. Anti-inflammatory effects of GCs are mediated by the interference of the GR with different transcription factors that play a critical role in controlling the expression of many proinflammatory genes. Many of them are under the positive control of AP-1 and/or NFκB, in such a way that GR antagonism with these transcription factors is believed to underlie most of the anti-inflammatory and immunosuppressive actions of GCs [21,38,39,40,41]. However, tethering of the monomer GR to many other transcription factors, such as IRFs, STATs CREB, GATA, and T-bet, has also been described to participate in the repression of inflammatory gene expression [21,27,40,60,62]. COX-2 transcriptional regulation depends on multiple transcription factors with different contributions depending on the stimuli and cell type analyzed [7,8,9,10,11]. Our studies with different promoter constructs show that the inhibitory effects of Dex on COX-2 transcriptional activation require the presence of two NFAT sites in the COX-2 promoter. Whereas other reports have shown the potential role of GR interference on NFκB to explain COX-2 downregulation by GCs in macrophages upon LPS activation [56], herein, we have identified the proximal region of COX-2 promoter as essential in Dex-mediated inhibition of COX-2 promoter induction by PMA + Ion in activated T cells. This region lacks the NFκB response elements of the COX-2 promoter, but contains two NFAT sites that are essential for the induction of COX-2 expression in T cells [13]. These results, together with the results obtained with the COX-2 promoter construct P2-431 NFκBmut, discard a relevant contribution of NFκB-mediated repression by GCs in this cell type. Likewise, other reports have discarded NFκB-mediated GC inhibition of gene expression in stimulated T cells, as in the case of IL2, IL4, and granulocyte monocyte colony stimulating factor (GMCSF) [42,43,63].

In addition, we have found that Dex was able to inhibit the increase in transcriptional activation of a NFAT-Luc reporter construct containing three tandem copies of the distal NFAT site of the human IL2 promoter. NFAT usually cooperates with components of the AP-1 family of transcription factors, which cooperatively bind with NFAT to composite DNA elements found in the promoter region of many NFAT target genes [46,64,65], determining a possible involvement of AP-1 in the actions exerted by Dex on NFAT-mediated transcription, as in the case of GC-mediated inhibition of GMCSF and IL2 transcription [43,44,64]. Nevertheless, Dex inhibited the transcriptional activation of IL4, IL13, as well as TNFα promoters, in which the regulation by NFAT plays a relevant role and has been described to be independent of AP-1 cooperation [46,47,48].

The activation of NFAT requires its dephosphorylation by calcineurin phosphatase and translocation to the nucleus, where it binds to NFAT response elements [66]. Our results indicate that Dex does not affect NFAT dephosphorylation, suggesting that the inhibitory mechanism on NFAT-mediated transrepression may occur after NFAT translocation to the nucleus, and could take place by GR–NFAT interaction in this cellular compartment [42,67]. In this regard, our results show that the overexpression of increasing quantities of NFATc2 reverses the inhibitory effect of Dex mediated by GRwt or GRLS7 receptors on the transcriptional activation of the COX-2 promoter, supporting the existence of a negative cross-talk between NFAT and GR.

Once in the nucleus, the activity of NFAT can be modulated at the level of their intrinsic ability of transactivate gene expression. NFATc2-dependent transcription is controlled by two different transactivation domains (TAD), at the N- and C-terminal ends of the protein. The TAD of the N-terminal region of this transcription factor plays a fundamental role in NFAT-mediated gene regulation [49,52,53]. Interestingly, Dex was able to interfere with the activity of a construct bearing the GAL4 DNA binding domain, which targets large heterologous proteins to the nucleus, fused to the N-terminal TAD of NFAT. PMA+Ion-mediated upregulation driven by the GAL4-NFATc2 (1–415) construct, in which the main regulatory sequences are present, was substantially inhibited by Dex, both in the presence of GRwt or GRLS7. These results provide additional evidence of Dex-mediated GR transrepression of NFAT activation at the nuclear level. Similar results have been reported for GAL4 constructs fused to the transactivation domain of the p65 subunit of NFκB, whose activity is repressed after Dex treatment [41]. Therefore, regulation of the activity of transactivation domains of different transcription factors could be another mechanism for the Dex regulation of gene transcriptional activation. The N-terminal TAD of NFATc2 is able to recruit co-activators, such as the p300/CBP, and is regulated by different signals mediated by PMA + Ion that induce phosphorylation of the N-terminal TAD of NFATc2, strongly enhancing the transcriptional activity of this factor [68,69]. A variety of kinases have been shown to be involved in the regulation of NFAT TAD activity that are essential for the activation of genes whose regulation depends on NFAT [54,55,70,71,72,73]. Therefore, additional mechanisms mediating GC-GR transrepression, such as competition with coactivators, as well as interference with kinase-activating cascades [22,74,75], could explain the interference with transcription factor TADs.

In summary, these results contribute to the understanding of the anti-inflammatory properties of GCs by their ability to interfere with the signal transduction pathways that lead to the activation of NFAT in T cells, thus inhibiting the induction of pro-inflammatory genes whose regulation is dependent on this transcription factor, such as COX-2.

## 4. Materials and Methods

### 4.1. Cell Culture and Reagents

The parental Jurkat human leukemic T cell line (Jurkat) and Jurkat cell lines stably transfected with GRwt (Jurkat GRwt) or the mutant version GRLS7 (Jurkat GRLS7) were kindly provided by Dr. C. Caelles (Institute for Research in Biomedicine, Universitat de Barcelona, Barcelona, Spain) [37]. Cells were cultured in RPMI medium (Invitrogen) supplemented with 10% foetal bovine serum (FBS) (BioWhittaker-Lonza, Basel, Switzerland) and 100 U/mL penicillin, 100 μg/mL streptomycin, 1000 U/mL gentamycin, 2 mM glutamine, and 0.1 mM non-essential amino acids. Purified human T cells were obtained from buffy coats of healthy donors by Ficoll-Paque Plus (GE healthcare, Chicago, IL, USA) centrifugation. The PBL fraction was plated, and adherent cells were removed. Purified T cells were obtained by passing the nonadherent population through a nylon fiber wool column, as previously described [12]. The purity of the population, detected by flow cytometry, was greater than 95% CD3^+^ cells. Studies with buffy coats from de-identified samples (without any direct or indirect personal identifiers) of healthy blood donors recruited by the Blood Transfusion Centre of the Comunidad de Madrid were approved by the Research Committee of Blood Transfusion Centre and the CSIC Research Ethics Committee in accordance with national and international guidelines.

Cells were stimulated with phorbol 12-myristate 13-acetate (PMA; Sigma-Aldrich-Merck, Madrid, Spain) at 15 ng/mL and/or A23187 calcium ionophore (Ion; Sigma-Aldrich-Merck) at 1 µM. The activation of human T cells through the TCR/CD3 complex and the CD28 receptor was conducted by adding the purified T cells to plates coated with anti-CD3 Ab (5 µg/mL), followed by the subsequent addition of anti-CD28 Ab (1 µg/mL) (Biolegend, San Diego, CA, USA). Dexamethasone (Dex; Sigma-Aldrich-Merck) (0.1 to 1 µM) and Cyclosporin A (CsA; Calbiochem, San Diego, CA, USA) (100 ng/mL) were added 1 h before the addition of PMA and Ion or anti-CD3/CD28.

### 4.2. Plasmid Constructs

The expression vectors pRc/βact, pRc/βact-GRwt, and pRc/βact-GRLS7 were provided by Dr. C. Caelles. pEFBOS-NFATc2 has previously been described [37,76]. pGRETK-Luc contains two GRE sites from the mouse mammary tumor virus (MMTV) fused to the thymidine kinase promoter. The plasmid TNFα-Luc contains a region 1311 bp upstream from the transcriptional initiation site of human TNFα promoter [77]. The reporter constructs, NFAT-Luc, containing three tandem copies of the NFAT binding site fused to the IL2 minimal promoter, and the IL2-Luc plasmid, which contains the region spanning from 2326 to 145 of the human IL2 promoter, have been described previously [78]. Both were a generous gift of Dr. G. Crabtree (Stanford Medical School, Stanford, CA, USA). The pGAL4-NFATc2 constructs contains either the first 1–104 or 1–415 amino acids of the N-terminal TAD of human NFATc2 fused to the DNA-binding domain of yeast GAL4 transcription factor [54]. The pGAL4-Luc reporter plasmid includes five GAL4-DNA binding sites fused to the luciferase gene [79]. pIL4-Luc and pIL13-Luc are reporter constructs driven by the promoter regions of the IL4 or IL13 genes [46,80]. The different COX-2 promoter Luc constructs (P2-1900 (−1796 to +104), P2-274 (−170 to +104), P2-150 (−46 to +104), and P2-431(−327 to +104) κB-mut) have previously been described [12,81].

### 4.3. mRNA Analysis

Total RNA (1 μg) was reverse-transcribed into cDNA and used for PCR amplification with either human COX-2-, IL2-, or GAPDH-specific primers [12]. Amplified cDNAs were separated by agarose gel electrophoresis and bands visualized by ethidium bromide staining. For quantitative real-time RT-PCR analysis, total RNA was reverse-transcribed using the components of the “High Capacity cDNA Archive Kit” (Applied Biosystems, Foster City, CA, USA). The amplification of the cDNAs was performed using the Taq Man Universal PCR Master Mix (Applied Biosystems) on an ABIPRISM7900HT instrument (Applied Biosystems) for 40 cycles with specific primers and Taqman probes: Hs00153133-m1 for *COX-2*, Hs00174114-m1 for *IL2*, Hs00174128-m1 for *TNFα*, and Hs99999901-m1 for *18S ribosomal RNA* (rRNA) (Applied Biosystems). All samples were run in triplicate and normalized by the expression of the endogenous *18S rRNA* gene. The quantification of gene expression by real-time RT-PCR was calculated by the comparative threshold cycle (ΔΔCT).

### 4.4. Immunoblot Analysis

Total protein extracts were obtained by lysis in Igepal buffer (50 mM Tris-HCl pH 8, 10 mM EDTA, 150 mM NaCl, 0.1% SDS, 1% Igepal) with protease inhibitors (aprotinin, leupeptin, and pepstatin at 10 µg/mL) and PMSF (phenyl-methylsulphonyl fluoride, 0.5 mM). The protein concentration was determined by the BCA method (Thermo Fisher Scientific, Waltham, MA, USA). Cell lysates were subjected to Western blot analysis using conventional SDS-PAGE gel electrophoresis and protein transfer to nitrocellulose filters. Extracts were separated on 6% (for the detection of NFATc2) and 10% (for COX-2 protein analysis) polyacrylamide gels (acrylamide/bis-acrylamide, 29:1). For the detection of COX-2, the membranes were incubated overnight at 4 °C with a monoclonal mouse anti human COX-2 antibody (Clone CX229; Cayman Chemical company, Ann Arbor, MI, USA) at 1:1000 dilution in blocking buffer. In the case of NFATc2, the membranes were incubated for 2 h at room temperature with the anti-human NFATc2 rabbit antiserum 672 (1:4000 dilution), raised against a peptide containing residues 53–70 of human NFATc2 (generous gift of Dr. J. M. Redondo, CNIC, Spain). The β-Actin levels were determined as a control of loading in each lane with a polyclonal goat anti-human specific antibody (sc1616, Santa Cruz Biotechnology Inc., Santa Cruz, CA, USA). The membranes were then incubated for 1 h with either rabbit anti-mouse, goat anti-rabbit (Thermo Fisher Scientific), or donkey anti-goat (Santa Cruz Biotechnology) IgGs secondary Abs linked to horseradish peroxidase at 1:15,000 dilution. The membranes were developed with the SuperSignal West Pico chemiluminescence system (Thermo Fisher Scientific). The scanned images of protein bands were quantified with the ImageJ program. The amount of COX-2 was normalized to β-actin levels. The ratio of phosphorylated vs. dephosphorylated NFATc2 was also determined by quantification of the respective bands.

### 4.5. Transfection and Luciferase Assays

The transcriptional activity was measured in transiently transfected Jurkat cells using luciferase reporter gene assays. Cells were transfected, unless otherwise indicated, with 0.5 µg of the different luciferase reporter constructs using Lipofectamine Plus, as recommended by the manufacturer (Invitrogen, Carlsbad, CA, USA). For transactivation assays, Jurkat cells were cotransfected with 0.5 µg of GAL4-DBD or GAL4-NFATc2 constructs and 0.25 µg of GAL4-Luc reporter plasmid.

In co-transfection experiments with NFATc2 or GR expression vectors, reporter constructs were transfected along with 0.1 to 1 µg of the indicated plasmid. The total amount of DNA in each transfection was kept constant by using the corresponding empty expression vectors. Transfected cells were treated with different stimuli as indicated. Then, cells were harvested and lysed, and luciferase activity was determined by using a luciferase assay kit (Promeg Biotech Ibérica, Madrid, Spain) in a luminometer, Monolight 2010 (Analytical Luminescence Laboratory, San Diego, CA, USA). The experiments were performed in triplicate and normalized by the mg of protein. The results are expressed as fold induction ± SEM (RLUs in the experimental samples/RLUs in the experimental controls).

### 4.6. Prostaglandin Measurement

Purified human T cells were collected after the different treatments and PGE_2_ or PGF2_α_ levels were measured in supernatants after cell incubation at 37 °C for 30 min with an excess of arachidonic acid (10 µM) in the presence or absence of the selective COX-2 inhibitor, NS398 (1 µM) (Cayman Chemical), in Hank’s balanced salt solution (HBSS). Samples were analyzed in triplicate by competitive PGE_2_ or PGF2_α_ immunoassay EIA kits, following the manufacturer’s instructions (Cayman Chemical).

### 4.7. Statistical Analysis

The results are shown as the mean ± SEM of triplicate determinations of a representative experiment of at least two independent ones. The differences between means were analyzed by one-way ANOVA with a post-hoc Tukey’s multiple comparison test. In all analyses, *p* < 0.05 was considered statistically significant. The data were analyzed using GraphPad Prism 5 software.

## Figures and Tables

**Figure 1 ijms-23-13275-f001:**
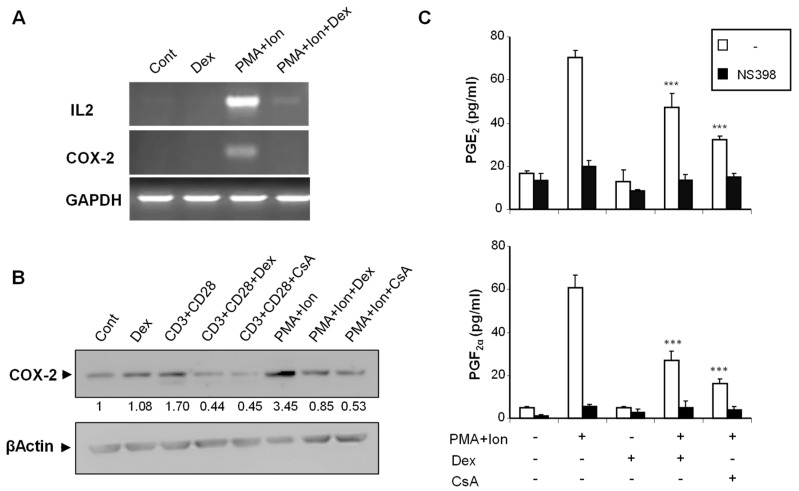
Dex inhibits COX-2 induction in activated human T cells. Purified human T cells were treated with PMA (15 ng/mL) plus Ca^++^ ionophore (Ion; 1 μM) or with anti-CD3/CD28 Abs (5 µg/mL; 1 µg/mL). Dex (1 µM) or CsA (100 ng/mL) was added 1 h before stimulation. (**A**) Levels of IL2, COX-2, and GAPDH mRNAs were determined by RT-PCR. An aliquot of the amplified DNA was separated on an agarose gel and stained with ethidium bromide for qualitative comparison. (**B**) COX-2 protein levels were analyzed by Western blot using anti-COX-2 mAb. β-Actin levels were determined as a control of loading. Numbers below the bands indicate COX-2 protein levels measured as the intensity of each band relative to β-Actin. (**C**) PGE2 and PGF2α were measured in supernatants of T cells treated with PMA + Ion, Dex, and CsA as indicated. PGs production in the presence or absence of the COX-2 inhibitor NS398 (1 μM) was determined by an EIA assay, as described in Materials and Methods. Results shown are from a representative of two independent experiments performed in triplicate, and are expressed as mean ± SEM. (*** *p* < 0.001; vs. PMA + Ion treatment).

**Figure 2 ijms-23-13275-f002:**
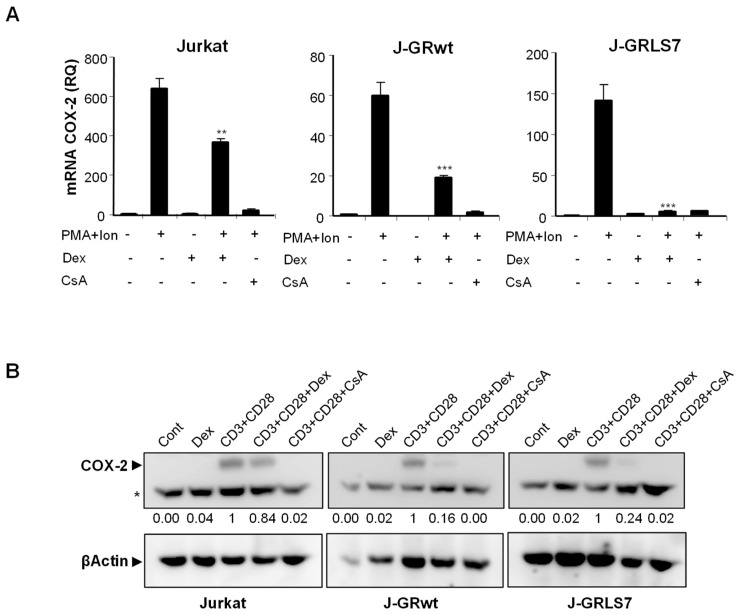
Effect of Dex on COX-2, IL2, and TNFα expression in activated Jurkat cells. (**A**) Analysis of COX-2 mRNA levels by quantitative real-time RT-PCR in Parental, J-GRwt, and J-GRLS7 Jurkat cells treated with PMA + Ion (15 ng/mL + 1 μM) or with anti-CD3/CD28 Abs (5 µg/mL; 1 µg/mL) for 18 h in the presence or absence of Dex (1 μM) or CsA (100 ng/mL). Results shown are from a representative of two independent experiments performed in triplicate and are expressed as RQ ± SEM (** *p* < 0.01; *** *p* < 0.001; vs. PMA + Ion treatment). (**B**) COX-2 protein levels were analyzed by Western blot in cell extracts. Nonspecific bands (marked by an asterisk) are also shown as internal loading control. Normalized relative densitometric quantification of COX-2 bands with respect to βActin loading control are shown below each lane.

**Figure 3 ijms-23-13275-f003:**
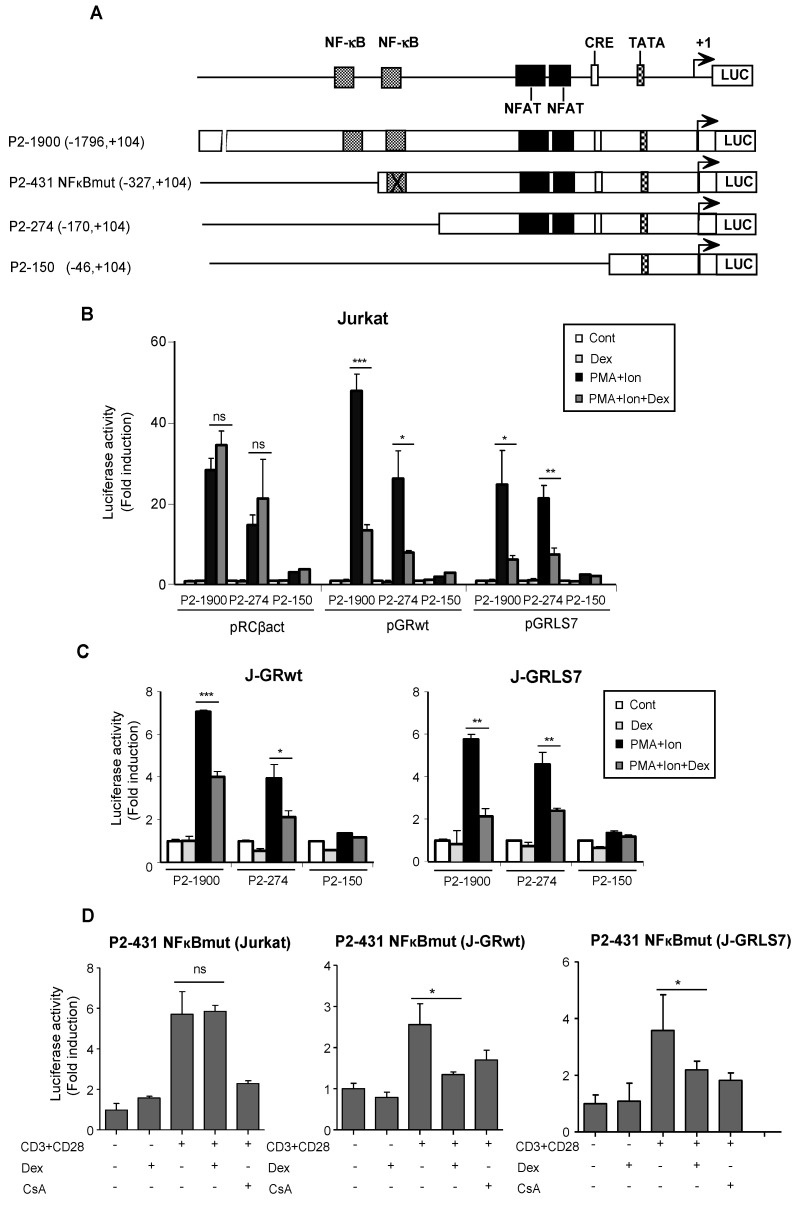
Dex effects on COX-2 promoter activity. (**A**) Schematic representation of the deletions ranging from −46 to −1796 bp relative to the transcription start site of the COX-2 promoter. Cis-acting consensus sequences are denoted by boxes. Jurkat cells were transfected with the indicated COX-2 promoter constructs and cultured in the absence (Cont) or presence of PMA + Ion or CD3/CD28 Abs for 18 h and assayed for luciferase activity. Dex (1 µM) was added 1 h before stimulation. (**B**) Cotransfection assays with COX-2 promoter constructs along with empty plasmid (pRCβact) or expression vectors for GR constructs wt or LS7 mutant. (**C**) Luciferase assays in Jurkat GRwt or GRLS7 cell lines transfected with the different COX-2 promoter deletions as indicated. Results shown are from a representative of two independent experiments performed in triplicate, and are expressed as fold induction over the unstimulated control samples (mean ± SEM). ns: non-significant; * *p* < 0.05; ** *p* < 0.01; *** *p* < 0.001; vs. PMA + Ion treatment. (**D**) Luciferase assays in Jurkat, J-GRwt, or J-GRLS7 cell lines transfected with the P2-431 NFκBmut COX-2 promoter construct. Results shown from a representative of two independent experiments performed in triplicate are expressed as fold induction mean ± SEM. ns: non-significant; * *p* < 0.05; vs. CD3/CD28 treatment.

**Figure 4 ijms-23-13275-f004:**
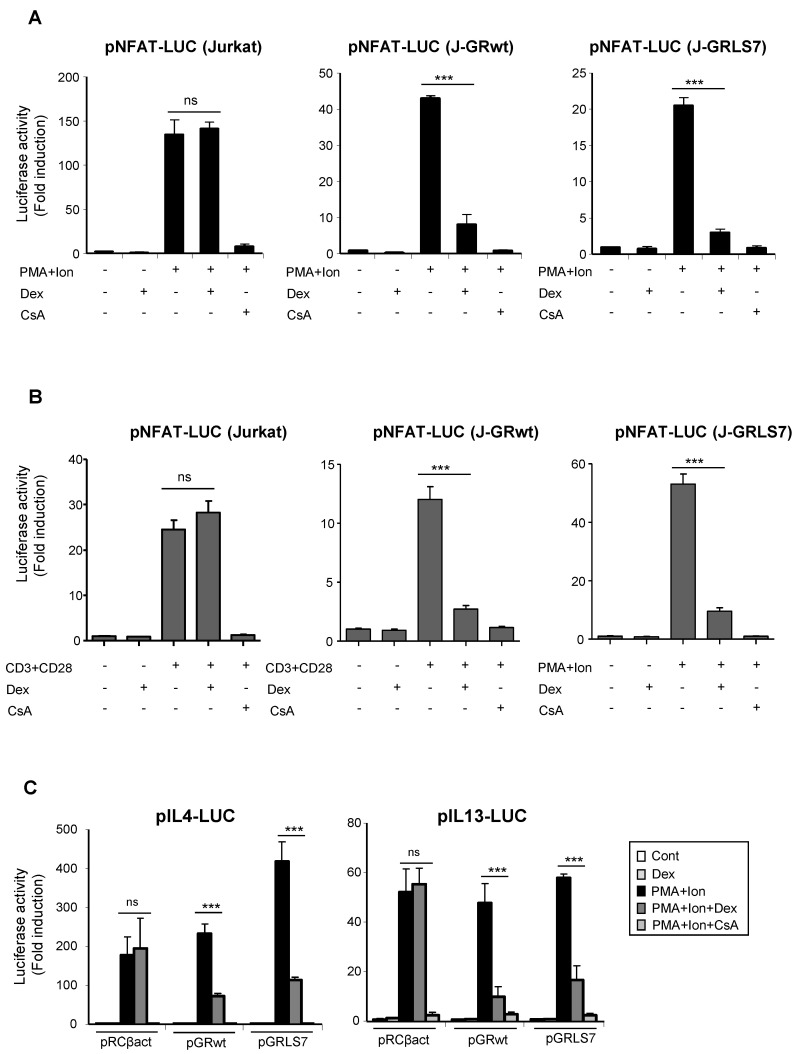
Dex reduces NFAT-mediated transcriptional activity in Jurkat T cells in the presence of GR or GRLS7. Transfected cells were pre-treated or not with Dex (1 µM) or CsA (100 ng/mL) before stimulation with PMA + Ion (15 ng/mL +1 μM) or with anti-CD3/CD28 Abs (5 µg/mL; 1 µg/mL). (**A**,**B**) Parental, J-GRwt, and J-GRLS7 Jurkat cells were transiently transfected with the luciferase reporter construct pNFAT-Luc and treated as indicated. Results shown are from a representative of three independent experiments performed in triplicate, and are expressed as fold induction mean ± SEM. ns: non-significant; *** *p* < 0.001; vs. PMA + Ion (in **A**) or CD3/CD28 treatment (in **B**). (**C**) Jurkat cells were co-transfected with luciferase reporter plasmids containing IL4 or IL13 promoter regions along with empty plasmid (pRCβact) or expression vectors for GRwt or GRLS7. Results are represented as fold induction (PMA + Ion over unstimulated control samples) (mean ± SEM). (*** *p* < 0.001 vs. PMA + Ion treatment). Results shown are from a representative of two independent experiments performed in triplicate and are expressed as fold induction mean ± SEM. ns: non-significant; *** *p* < 0.001; vs. PMA + Ion treatment.

**Figure 5 ijms-23-13275-f005:**
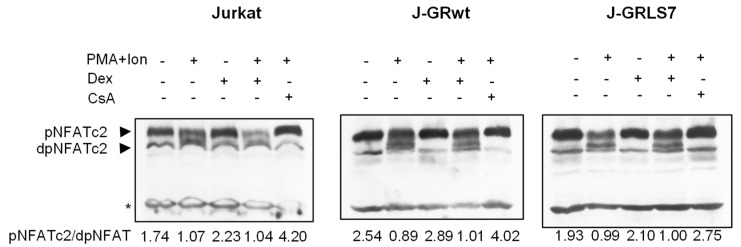
Dex treatment does not disturb NFAT dephosphorylation. Western blot analysis of NFATc2 in total protein extracts from parental, J-GRwt, or J-GRLS7 Jurkat cells pre-treated or not with Dex (1 µM) or CsA (100 ng/mL) before stimulation with PMA +Ion for 90 min. Bands of phosphorylated (p-) and dephosphorylated (dp-) NFATc2 are indicated by arrows. Nonspecific bands (marked by an asterisk) are also shown as internal loading control. The ratio of phosphorylated vs. dephosphorylated NFATc2 was determined by quantification of the respective bands and it is shown under each lane.

**Figure 6 ijms-23-13275-f006:**
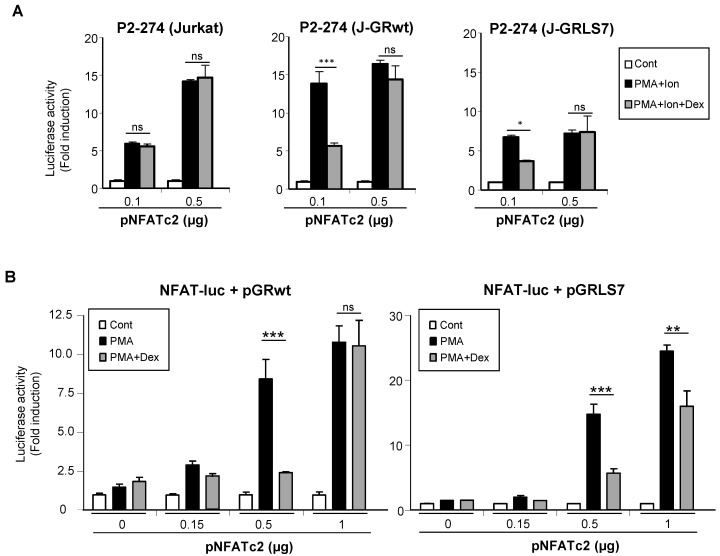
Negative cross-talk between GR- and NFAT-mediated transcription. (**A**) Parental, J-GRwt, and J-GRLS7 Jurkat cells were transiently transfected with the COX-2 promoter construct P2-274-Luc together with increasing quantities of NFATc2 expression plasmid as indicated. Cells were treated for 1 h with Dex (1 μM) before O/N PMA + Ion (15 ng/mL + 1 μM) stimulation. (**B**) Jurkat cells were transiently transfected with the NFAT reporter construct pNFAT-Luc together with pGRwt or pGRLS7 expression plasmids along with increasing quantities of an NFATc2 expression plasmid (µg). Cells were treated for 1 h with Dex (1 μM) before O/N PMA (15 ng/mL) treatment. After transfection, cells were lysed, and luciferase activity was determined. Results shown are from a representative of two independent experiments performed in triplicate, and are expressed as fold induction mean of PMA + Ion or PMA samples over unstimulated control ones (mean ± SEM). ns: non-significant; * *p* < 0.05; ** *p* < 0.01; *** *p* < 0.001; vs. PMA + Ion or PMA treatment.

**Figure 7 ijms-23-13275-f007:**
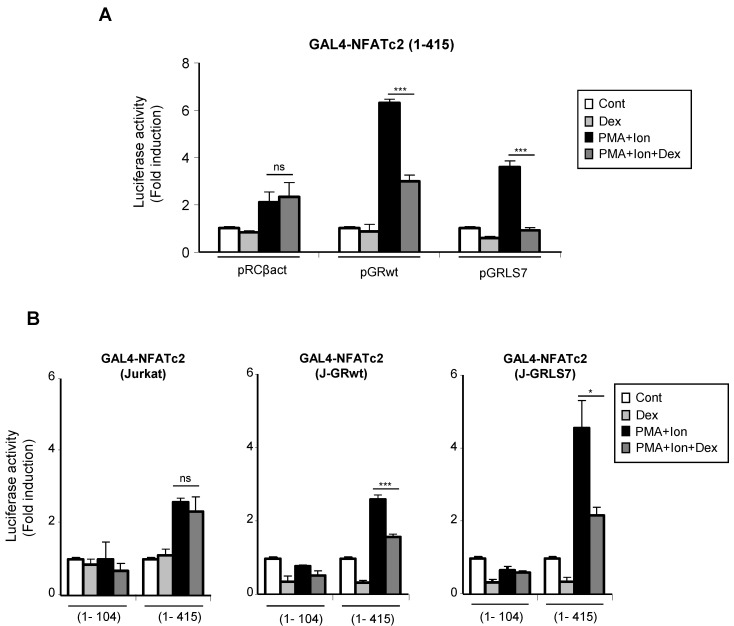
Dex interferes with the transactivating activity mediated by the N-terminal TAD of NFAT. (**A**) Jurkat cells were co-transfected with a pGAL4-Luc reporter plasmid and the GAL4-NFATc2 (1–415) expression vector containing the N-terminal TAD of NFATc2, along with empty plasmid (pRCβact) or expression vectors for GR constructs wt or LS7 mutant. (**B**) Parental, J-GRwt, and J-GRLS7 Jurkat cells were transiently transfected with the pGAL4-Luc reporter plasmid and the GAL4-NFATc2 constructs expressing amino acids 1–104 or 1–415 of the N-terminal TAD of NFATc2. Transfected cells were pre-treated or not with Dex (1 µM) before stimulation with PMA + Ion (15 ng/mL + 1 µM) for 18 h. Luciferase activity is represented as fold induction (PMA + Ion over unstimulated control samples) (mean ± SEM). ns: non-significant; * *p* < 0.05; *** *p* < 0.001; PMA + Ion + Dex vs. PMA + Ion treatment). Results shown are from a representative of three independent experiments performed in triplicate.

## Data Availability

Data supporting the reported results used and/or analyzed during the current study are available from the corresponding author (miguelangel.inniguez@uam.es) upon reasonable request.

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
