# Peer review of "Regulation of Cyclooxygenase-2 Expression in Human T Cells by Glucocorticoid Receptor-Mediated Transrepression of Nuclear Factor of Activated T Cells"

_ijms, 2022, doi:10.3390/ijms232113275_

Round 1

Reviewer 1 Report

Glucocorticoids (GCs) are potent anti-inflammatory and immunosuppressive agents commonly used in the treatment of inflammatory and autoimmune diseases. Transcriptional induction of gene expression by GCs depends on ligand activated Glucocorticoid Receptor binding to glucocorticoid response elements in the promoter region of target genes.

In this manuscript, Cacheiro-Llaguno et.al. present work showing that the COX-2 expression is regulated by Glucocorticoid Receptor mediated by a transrespression NFAT. They conclude that Dexamethasone, a synthetic GC, treatment did not disturbed NFAT dephosphorylation but interfered with activation mediated by the N-terminal transactivation domain (TAD) of NFAT, thus pointing to a negative cross-talk between Glucocorticoid Receptor and NFAT at the nuclear level. Furthermore, they exclude NFκB.

The manuscript is generally well written, however there are major concerns regarding the stimulation of the T lymphocytes and the methods used to conclude their results.

I have three major concerns:

1.) In all experiments PMA/Ionomycin was used to activate primary human or Jurkat T Lymphocytes. A combination of PMA/Iono is leading to a very strong/unphysiological response with no surface receptor stimulation and that is how T lymphocytes naturally get activated. PMA is a small organic compound which diffuses through the cell membrane and directly activates Protein Kinase C, and Ionomycin, a is calcium ionophor, is triggering a Ca2+ response. To have a more physiological stimulation either beads coated with anti-CD3/CD28 antibodies or soluble anti-CD3/CD28 antibodies must be used.

2.)A lot of the experiments are done using transfection and luciferase assay. Transfections of one or two constructs can lead to numerous downstream effects furthermore the results from the luciferase assay have a huge variation (See for example Fig4A, FigS3). One problem is that the y-axis are not in the same range which makes the comparison not really possible. Furthermore, it would be a lot more convincing if further methods like immunofluorescence staining for NFAT, FACS or ELISA methods for IL2, IL4, IL13 and TNFa would be applied. These method could also be applied to primary human T cells.

3.) From the result in Fig3 you conclude that NFκB is not contributing to the repression by GC (P 12 Line 352-353) this it not convincing since you already see a reduction if you compare the P2-1900 to the P2-274, so it seems that NFκB is involved. To convincingly show that NFκB is not involved a construct just lacking the NFAT binding site has to be generated.

MAJOR CRITIQUES

1.         I would move the first part of the discussion (line 292-306) to the introduction part this would help the reader a lot.

2.         Figure 1B, 2B and 5: For all the WBs loading controls are missing eg. GAPDH or Ponceau red.

3.         Figure 1C: Why is there PGE2 production if there is no COX expression after treatment with PMA-Ion + Dex/ PMA-Ion + CsA?

5.         Figure 4 the title should be changed to “DEX reduces NFAT-mediated transcriptional activity in Jurkat T cells if GR or GRLS7 is expressed”. Fig4A why is the y-axis so different? Not comparable in this presentation. Furthermore, instead of overexpressing a pNFAT-Luc construct for example IF staining of NFAT should be used.

6.         Discussion Line 327-328: You conclude that: ”This mode of action relies on the interference with the activity of transcription factors by protein–protein interaction with the GR” this is not shown in the manuscript. This should be done for example with a Co-IP.

MINOR POINTS

P2 Line 70: “intracellular receptor (GR)” please insert “glucocorticoid”

P2 Line 96: I would just shortly explain the function of the LS7 mutant here.

P3 Line 99-101: “GRE-mediated transactivation” please introduce the abbreviation in line99 it is done in line 101

P11 Line 312 and 332: “in vivo” should be italics “in vivo

P12 Line 356: “containing” is double, delete one

Author Response

Pleasesee the attachment

Reviewer 2 Report

The authors study the mechanism of action by which a glucocorticoid, dexamethasone, inhibits T cell functions using in vitro experiments on primary human T cells and human Jurkat T cell lines. They carefully address the mechanism of action of the receptors DNA binding function, and promoter elements. They find Dex did not require DNA binding domain to inhibit COX-2 transcription. Dex interfered with NFAT function (but not its phosphorylation), which may explain how it attenuated the COX-2 transcription. The discussion was very clear and did not overstate the claims.
The transfection experiments were very well controlled, with appropriate methodology used, and they used mutant GR and empty vector as comparison and control. The GRE luciferase assay was normalized to protein in mg, and expressed as fold change. The Western blots have some issues that could be addressed to make them more convincing . The manuscript has some minor typos, and clarifications would improve it.

minor issues

line 26  disturbed

line 55 early after T cell receptor triggering > can be more specific, seconds, minutes hours?

line 69 extra bracket

line 80 As consequence

line 97 its not clear if Parental Jurkat T cells data was from reference 36, or the present manuscript.

line 220 , line 273 typo  o .. or

line 420 the 1mM ionomycin sounds high it would probably kill the cells, but in the figure it says 1 micromolar which is more normal, perhaps a typo in methods section? Authors should check all the units of measurement

line 458 specify what % gel was used, and acrylamide ratio.

Specify what dilution of primary antibodies used....What was the secondary antibody, and its dilution, and company.

line 448 TaqMan probes should be specified

Explain what 'de-indentified' means regarding the human blood samples, did the blood bank remove personal identifiers from the sample.

Figure 4, in the key two bars are white. The shades of grey are very close, could use more contrast.

major issues

Western Blots, more of the Western blot membrane should be shown in the manuscript figures for reader to assess quality and see background bands (unless the reader will have access to the raw data that was made available to the reviewers). A strip/reblot  of housekeeper protein should be shown as a protein loading control. If its not possible now, then a ponceau S stain or a coomassie could confirm equal loading, especially in the lanes without any COX2 signals. There is a low quality Figure 2b blot that is not convincing.
In the additional raw data file, the whole blot was shown, but the numbers on molecular weight markers were mostly missing, just 70kDa shown. One blot had a blocking issue perhaps, Figure 3 middle, shown in the additional raw data, also appears very exposed... in the manuscript, figure 3 does not have western blot, but in the additional data the label says Figure 3 (it probably meant to say figure 2 in the additional file).  Would suggest redoing this blot because the band is not clean enough, and fixing all the labelling issues on the additional data.

Figure 1B primary cells, the dex effect was weak, although it was significant. Can authors comment in discussion, was a complete block of PGE2 expected? In Figure 2, the COX2 is strongly inhibited by dex in Jurkat, so why is PGE2 still being detected. Or does the drug work differently on primary versus Jurkat cells.

Figure 5 would be more convincing with a quantification and statistical analysis of the % phosphorylation

The statistics were done either by pooling data from at least two experiments, or showing one experiment as a representative. However, it is not clear which figures were pooled and which were representative. And which experiments were done twice, which were done three times. Suggest adding this to each caption if its different for each figure.

Bonferroni's multiple comparison test was made, the methods say P < 0.05 was significant, although figures show additional  levels of significance 0.01, 0.001. However, in Bonferroni's test the alpha level is adjusted by the number of tests being made. alpha/N  thus p < 0.05 /N.  Suggest using APA guidelines, for example, if N = 3, it would be written as '... adjusted alpha level of .016 per test (.05/3).

Round 2

Reviewer 1 Report

After the revision most of the raised points were addressed or discussed.

There are just a few minor spell check errors:

P2 Line 55: “early as 1 h. after T cell” please remove the dot

P3 Line 108: “expression vec-tors were also performed” please remove the “-“

Sometimes “T cells” as well as “T lymphocytes” are used. I would suggest to just decide for either one and use only one form throughout the manuscript.

Author Response

We would like to thank the reviewers for taking the necessary time and effort to review the manuscript. We sincerely appreciate all your valuable comments and suggestions.

Response to “minor spell check errors”

We have corrected two minor typos as indicated, as well as decided to use “T cells” instead of “T lymphocytes” throughout the manuscript, as suggested.

P2 Line 55: “early as 1 h. after T cell” please remove the dot

Done

P3 Line 108: “expression vec-tors were also performed” please remove the “-“

Done

Sometimes “T cells” as well as “T lymphocytes” are used. I would suggest to just decide for either one and use only one form throughout the manuscript.

“T lymphocytes” has been changed to “T cells”

Reviewer 2 Report

The manuscript was adequately revised. In particular, the western blot data and statistical analysis is convincing. 

Author Response

We would like to thank the reviewers for taking the necessary time and effort to review the manuscript. We sincerely appreciate all your valuable comments and suggestions.